# Transcriptome Analysis of the Immortal Human Keratinocyte HaCaT Cell Line Damaged by Tritiated Water

**DOI:** 10.3390/biology12030405

**Published:** 2023-03-03

**Authors:** Yan Zhang, Yuanyuan Zhou, Hui Wu, Zhuna Yan, Jinwu Chen, Wencheng Song

**Affiliations:** 1Anhui Province Key Laboratory of Medical Physics and Technology, Institute of Health & Medical Technology, Hefei Institutes of Physical Science, Chinese Academy of Sciences, Hefei 230031, China; 2School of Life Science, Hefei Normal University, Hefei 230061, China; 3Hefei Cancer Hospital, Chinese Academy of Sciences, Hefei 230031, China; 4Collaborative Innovation Center of Radiation Medicine of Jiangsu Higher Education Institutions and School for Radiological and Interdisciplinary Sciences, Soochow University, Suzhou 215123, China

**Keywords:** tritiated water, transcriptome, HacaT cells, cell viability

## Abstract

**Simple Summary:**

Tritium is one of the most abundant radioactive elements in nuclear waste and is difficult to remove. In addition, tritiated water can enter an organism through the skin, respiratory and digestive systems. Tritiated water damages a large portion of organs or even causes cancer due to internal radiation. In our study, the changes in the cell viability of the immortal human keratinocyte HaCaT cell line after exposure to tritiated water were investigated, and the related molecular mechanisms were analyzed using sequencing technology and bioinformatics methods. Meanwhile, a Western blot assay was conducted to verify some of the sequencing results. The results provide a theoretical basis for researching the mechanisms of tritiated water hazards.

**Abstract:**

Radioactive elements, such as tritium, have been released into the ocean in large quantities as a result of the reactor leakage accident. In this study, an MTT assay demonstrated that the viability of HacaT cells decreased after tritiated water treatment. Bioinformatics analysis was used to analyze gene changes in the HacaT cells. The sequencing results showed 267 significantly differentially expressed genes (DEGs), and GO enrichment analysis showed that the DEGs were mainly divided into three parts. The KEGG pathway analysis showed that the up-regulated DEGs were involved in Wnt and other pathways, while the down-regulated DEGs were involved in Jak–STAT and others. A Western blot assay was used to verify the parts of the sequencing results. This study was the first to explore the mechanism of tritiated water on HacaT cells using Transcriptome analysis. The results will provide a theoretical basis for the study of tritiated water hazard mechanisms.

## 1. Introduction

With the development of industry, the development and utilization of nuclear energy increased correspondingly, and a large number of nuclear power stations were built. Earthquakes, tsunamis and improper operation due to worker behavior all may lead to nuclear reactor leakage accidents, resulting in the release of large amounts of nuclear wastewater into the environment [1]. Generally, nuclear wastewater contains a large number of radioactive substances, among which tritium is the most common substance and the most difficult to remove [2]. Tritium is a radioactive isotope of Hydrogen, which mainly exists in three main forms: tritiated water, tritiated gas and organically bound tritium. Among them, the average energy of tritium *β*-rays is 5.7 keV, and the maximum is 18.6 keV. It has a range of about 0.56 μm in water, which is much smaller than the average diameter of the cell (10–20 μm) [3]; therefore, tritiated water can produce a biological effect on an organism through irradiation [4]. In addition, tritiated water is similar to H_2_O, which is volatile, forming tritiated water vapor. Tritiated water enters the body through the respiratory system, skin infiltration or the food chain, and it can be distributed in the human body through the blood circulation system in about 2–3 h [5]. 

Low doses of tritiated water may not cause damage to organisms or may even have protective effects on organisms. Stuart et al. found that frogs in Duke Swamp exposed to tritiated water radiation (5000–35,000 Bq/L) for a long time were able to reduce their susceptibility to radiation damage [6], which suggested an adaptive response in frogs exposed to radiation from tritiated water. Arcanjo et al. found that 0.4 and 4 mGy/h tritiated water exposure enhanced the expression of some genes related to DNA repair (h2afx and ddb2) [7]. Stuarta et al. found that when B-lymphoblast cells (3B11 and FHMT-W1) were exposed to 10–100 Bq/L tritiated water, intracellular ATP content increased, and cell vitality did not decline [8].

However, high doses of tritiated water radiation can damage organisms. An analysis of the chromosomes of nuclear workers exposed to tritium (average dose 9.33 mGy) in the UK found that the workers had an increased rate of chromosomal aberrations [9]. In addition, tritiated water leads to liver, ovarian, leukemia, lung, breast damage, skin cancer and osteosarcoma in rats [10,11,12]. For example, Yin et al. proved that tritiated water above 8 × 10^10^ Bq could induce liver cancer in male rats, but only above 3.7 × 10^12^ Bq, could it induce the occurrence of ovarian cancer in female rats [10]. Seyama et al. injected 7.4 × 10^8^ Bq of tritiated water into mice, which induced leukemia and lymphoma in more than 80% of the mice, as well as increasing the risk of lung cancer [11]. Balonov et al. found that exposure to tritiated water in rats for 6 months at 3.7 × 10^4^ Bq·g^−1^/Kg induced skin cancer and osteosarcoma [12]. In addition, tritium radiation damaged the cardiovascular [13,14,15], immune [16,17] and reproductive [18] systems. Li et al. found that when human B lymphocytes (AHH-1-1) were treated with 3.7 × 10^9^ Bq/L tritiated water for 48 h, cell apoptosis could be observed [17]. Lee et al. proved that micronucleus appeared in the blood samples of rats exposed to tritiated water after 14 d in rats with a total oral dose of 3.7 × 10^4^ Bq, and the appearance of micronucleus, which represents chromosome aberration. Nowosielska et al. treated mice with tritiated water of 0.888, 8.88 or 88.8 × 10^12^ Bq, and found that after 8 d, NK lymphocytes were damaged and that tritiated water promoted increased NO by macrophages [16]. Kamiguchi et al. found that after treating human sperm with 5.66–59.91 × 10^5^ Bq/mL tritiated water for 80 min, the incidence of chromosome aberration in the sperm structure increased linearly in a dose-dependent manner [18]. Along with some behavioral experiments that showed that the learning and memory ability of mice decreased significantly when they were treated with tritiated water above 24.09 × 10^4^ Bq/g tritiated water after 10 d [19]. Among them, studies have shown that high doses of tritium radiation also had a negative impact on the survival of algae, bacteria, shellfish and fish [20,21,22,23]. For example, Selivanova et al. proved that when the concentration reached 1 × 10^14^ Bq/L, tritiated water would damage bacterial cells [20]. Réty found that the cell density of the algal decreased significantly under 5.9 × 10^16^ Bq/L tritiated water [22].

Recently, some studies have focused on the mechanisms related to the decrease in cell viability caused by tritiated water. Quan et al. found that after incubation with 2 × 10^10^ Bq/L tritiated water for 3 h, the proportion of S-phase cells in human mesenchymal stem cells increased, while that of G1- and G2-phase cells decreased [24]. Vorob’eva et al. proved that Tritium could cause DNA damage by increasing the expression of *γ*H2AX and dsDNA [25]. In addition, Yan et al. used 3.7 × 10^6^ Bq/L tritiated water to induce the cell senescence of vascular endothelial cells [14], and Cui et al. demonstrated that this might be due to tritiated water down-regulating the expression of c-myc via miR-34a [13]. Qiu et al. found that treating rat nerve cells with 3.7 × 10^10^ Bq/L tritiated water for 8 h reduced the expression of neural cell adhesion molecule (NCAM), which could decrease the migration ability of nerve cells [26]. Zhou et al. treated mouse brain cells with tritium *β*-rays at doses of 0.19 Gy and above, which resulted in the increased gene expression of P53 in the brain cells [27]. 

In addition, since the skin is the largest tissue in the human body, tritiated water can easily reach the skin. Therefore, some studies have shown that tritiated water produced toxic effects on human skin cells. For example, Little et al. used a total dose of 100 Gy tritiated water to culture human skin fibroblasts for 100 h, which caused cell death [28]. However, the molecular mechanism of damage to epidermal cells caused by tritiated water has not been elucidated. Therefore, this study focused on the decreased cell vitality caused by tritiated water to the immortal human keratinocyte HaCaT cell line and related mechanisms. 

Next-generation RNA sequencing (RNA–SEQ) is an emerging, rapid and efficient method for gene expression analysis [29]. In this study, HacaT cells were used as experimental subjects, and transcriptome sequencing technology, as well as bioinformatics analysis, were used to explore the effects and mechanisms of tritiated water exposure on the HacaT cells, and Western blot was used to verify parts of the sequencing results, This study will provide a theoretical basis for the studying the mechanisms of high-dose tritiated water hazards.

## 2. Materials and Methods

### 2.1. Cell Line Culture

The immortal human keratinocyte HaCaT cell line (HaCaT) cells were purchased from keyGEN BioTECH (Cas: 20210513) and cultured in a complete medium composed of a DMEM cell medium (GIBCO, Carlsbad, CA, USA) supplemented with a 10% fetal calf serum (LONSERA, Shanghai, China) and 1% penicillin/streptomycin (NCM Biotech, Suzhou, China). The cells were grown in 35 mm dishes under standard cell culture conditions (37 °C and 5% CO_2_ in an incubator). 

### 2.2. Cell Viability Assay

HacaT cells were grown to 70% of the culture surface area after the cells were attached to the wall and fully stretched and exposed to different concentrations of tritiated water, and then the HaCaT cells were cultured for 48 h in a cell incubator. The cell viability was conducted using MTT (Sigma–Aldrich, St. Louis, MO, USA), and the absorbance (OD value) was measured at 492 nm. Then, the cell viability value was calculated on the basis of the following formula, as previously used [30].
(1)$$\mathrm{cell}\;\mathrm{viability}=\frac{{\mathrm{absorbance}}_{\mathrm{treatment}}}{{\mathrm{absorbance}}_{\mathrm{BK}}}\times 100\%$$

In this formula, absorbance_traeatment_ represents the OD value of the DZ and GJL group, and absorbance _BK_ represents that of the BK group. 

### 2.3. Radiation Treatment

The initial specific radioactivity of the tritiated water used in the experiment was 3.7 × 10^10^ Bq/L, which was obtained from the Institute of Nuclear Energy Safety (Hefei Institutes of Physical Science, CAS, Hefei, China). Tritiated water was filtered and sterilized before experimental use, and DMEM medium was used to dilute different concentrations of tritiated water. HacaT cells were treated with 0 Bq/L and 3.7 × 10^9^ Bq/L radiation as a blank group (KB), a control group (DZ) and a treatment group (GJL), respectively. Specifically, the cells in the KB group were cultured in 5 mL DMEM; the DZ group was cultured in 4500 μL DMEM and 500 μL sterile water, which was replaced with the same volume of tritiated water in the GJL group. Additionally, a follow-up experiment was conducted after 48 h of continuous exposure to tritiated water.

### 2.4. RNA Extraction

The DZ and GJL group cells (three replicates for each group) were selected for the subsequent experiments. The HacaT cells were treated with 0 Bq/L and 3.7 × 10^9^ Bq/L radiation. After being cultured for 48 h, the cells were digested by trypsin, collected into frozen storage tubes, and stored at −80 °C. After that, the total RNA from the HacaT cells was extracted according to the manufacturer’s instructions by using the TRIZOL kit (Invitrogen, Carlsbad, CA, USA). The quality of the RNA was evaluated by bioanalyzer Agilent 2100 (Santa Clara, CA, USA). The qualified RNA was used for subsequent experiments.

### 2.5. cDNA Library Preparation and Sequencing

The traditional mRNA enrichment method was used to extract the mRNA from the total RNA [31], which enriched the mRNA with polyA tail using magnetic beads with Oligo (dT). The RNA was fragmented with a Frag/Prime Buffer of about 200 bases and reverse transcribed with random N6 primers. Later, dsDNA was synthesized with the end flattened and phosphorylated at the 5′ end, with a bubblelike joint protruding a ‘T’ at the 3′ end. Then, they were amplified using PCR. After that, single-stranded DNA was formed and cycled to a single-stranded cDNA library. The constructed library was inspected, and the DNBSEQ platform was used for sequencing.

### 2.6. Sequencing Data Filtering and Reference Genome Comparisons

This experiment was commissioned by The Beijing Genomics Institute (BGI), and the specific steps were as follows: SOAPnuke software (V1.5.2) [32] was used for filtering to obtain high-quality sequencing data, i.e., clean reads. The readings containing adapters in sequencing fragments and an unknown base N content greater than 5% and the low–quality reads were removed from the sequencing fragment to obtain clean reads. HISAT2 2.0.4 was used to align the clean reads to the reference genome sequence [33], and Bowtie2 software was used for the Quality Control of the alignment.

### 2.7. Bioinformatics Analysis

To quantitatively analyze the DEGs of the two samples, the DEGSeq R package was used to analyze the differential expression based on the gene expression levels (principal components, correlations, differential gene screening, etc.). Furthermore, we set |log2FC| ≥ 0.6, Qvalue ≤ 0.05 screening DEGs between two samples, and then, the Gene Ontology (GO) and Kyoto Encyclopedia of Genes and Genomes (KEGG) databases were used to conduct an analysis of the sequencing results. 

### 2.8. Western Blot Verification

The HacaT cells were treated with 0 Bq/L and 3.7 × 10^9^ Bq/L radiation. After 48 h culture, the DZ and GJL group cells were lysed by an RIPA cell lysis buffer (Beyotime, Shanghai, China), and the proteins of the cells were extracted following the manufacturer’s instructions. Protein supernatants were separated by 10% SDS–PAGE gels and then transferred to the NC membrane (Beyotime, Shanghai, China), sealed with non-fat milk, and probed with primary antibodies at 4 °C overnight and incubated with HRP-conjugated secondary antibody for 40 min. Several antibodies were used as follows: anti–Wnt7b, anti–Jak, anti–STAT3 and anti–*β*–Actin (ABclonal, Wuhan, China). Later, ECL Plus Reagent (Thermo Fisher Scientific, Waltham, MA, USA) was used to develop, and Chemiluminescence Gel Imaging System (Tanon, Shanghai, China) was used to visualize the protein bands.

### 2.9. Statistical Analysis

The data of the assays were presented as means ± S.D. The data were determined using Student’s *t*-test. A *p*-value < 0.05 (*) was defined as statistically significant. 

## 3. Results

### 3.1. Tritiated Water Impaired the Viability of HacaT Cell

The results of the MTT experiment found that tritiated water could lead to the viability of HacaT cells significantly decreasing, as shown in Figure 1. After 48 h of radiation, the average OD value of the DZ group in the MTT experiment was about 0.94, while that of the treatment group was about 0.82, which was 13.0% lower than that of the DZ group. Thus, tritiated water could impair the viability of HacaT cells significantly.

### 3.2. Overview on Transcriptome Sequencing Data Quality and Difference Analysis of HacaT Cells Exposed to Tritiated Water

In this experiment, the BGISEQ platform was used to measure a total of six samples, and the samples from each group produced 6.66 G data on average. The average rate of genome alignment was 92.64%, and that of the gene set was 80.46%. A total of 15,807 genes were detected. The bases with sequencing Q20 accounted for more than 97.54% of all bases, while the bases with sequencing Q30, which accounted for more than 93.55% of all bases. In this case, the average sequencing efficiency was about 93.84%. Therefore, the sequencing data were of good quality and met the requirements for subsequent analysis (Table 1). In addition, Pearson’s correlation coefficient between every two samples was calculated in order to reflect the correlation of gene expression between the samples, which was presented in the form of a heat map. The higher the correlation coefficient, the more similar the level of gene expression, as shown in Figure 2. In this study, the square of Pearson’s correlation coefficient was all greater than 0.95. These outcomes indicated that the sequencing results were valid. 

In addition, from the DEGseq analysis, a total of 267 DEGs, 191 significantly up-regulated DEGs, and 76 down-regulated DEGs, are shown in Figure 3 (|log2FC| ≥ 0.6, Qvalue ≤ 0.05). Additionally, there are several outstanding DEGs on both sides and top. There are several outstanding DEGs on both sides and top, we included detailed information about these genes in the Appendix A. 

### 3.3. GO Enrichment Analysis of DEGs

GO enrichment analysis was divided into three parts: molecular function, biological process and cellular component. Figure 4 shows the GO enrichment analysis of the total DEGs between the GJL and DZ groups in terms of the biological process module; DEGs were enriched in cellular processes, biological regulation, metabolic processes, the regulation of biological processes and so on. In terms of the cellular component, DEGs were enriched in the cell, cell part, organelle, membrane and so on. While in terms of the binding, catalytic activity, molecular function regulator, molecular transducer activity and so on. Specifically, we list 10 of the most highly enriched signaling pathways according to their *p*-values in Table 2. In addition, all of the specific genetic information of the DEGSs analyzed by GO is listed in detail. (Appendix A.)

### 3.4. KEGG Pathway Enrichment Analysis

KEGG pathway enrichment analysis also performed on the DEGs in the DZ and GJL groups is shown in Figure 5. The up-regulated DEGs were mainly enriched in the neuroactive ligand-receptor interaction, amoebiasis, axon guidance, proteoglycans in cancer, Wnt and MAPK signaling pathway; while the down-regulated DEGs were mainly enriched in chemical carcinogenesis, fatty acid elongation, fatty acid metabolism, Jak-STAT signaling pathway and so on.

### 3.5. Tritiated Water Induced Protein Expression Differences in HacaT Cell

The results of the KEGG enrichment analysis showed that the expression of the Wnt signaling pathway was up-regulated, while Jak-STAT was down-regulated. Therefore, some genes (Wnt7b, Jak, STAT3) concluded that those two pathways were selected to verify the accuracy of the sequencing results of the Western blot assay. Figure 6 and Appendix A show the protein expression of the DZ and GJL groups; the expression of Wnt7b was significantly increased, while Jak and STAT3 decreased in HacaT cells exposed to tritiated water. The results were consistent with the transcriptome results, which proved the accuracy of the sequencing results.

## 4. Discussion

In this study, HaCaT cells were used as the experimental objects to explore the damage effects of tritiated water. We found a significant decrease in cell viability after 48 h of treatment with 3.7 × 10^9^ Bq/L tritiated water. Some previous reports in the literature proved that tritiated water could also inhibit the vitality of Human Umbilical Vein Endothelial Cells and cause cell senescence after long–term exposure [13,14,15]. In addition, tritiated water could also affect the vitality of rat lymphocytes and NK cells in terms of the immune system [16,17]. As a radioactive element, the damage to cells of tritiated water mainly included the following two aspects: first, tritium *β*–decay released energy, resulting in DNA single-strand break (SSB) or double-strand break (DSB), leading to cell apoptosis or cell senescence [34,35]. Some characteristics related to apoptosis and senescence ensued, such as cell cycle arrest, increased *γ*H2AX and IL-8 contents, as well as an increase in the proportion of positive cells stained with *β*-galactosidase (SA-*β*-gal) [14,24]. Second, as ionizing radiation, Tritium caused the radiolysis of water, which promoted the formation of ROS [36] to attack intracellular biological macromolecules such as DNA, protein and lipids, causing cell damage [37]. In addition, Li et al. treated AHH-1-1 with 3.7 × 10^6^ Bq/mL tritiated water, and only 72.1% of the cells survived [17]. We have similar experimental conditions; the HacaT cell survival rate was about 87% after treatment. This may be due to different cell types, resulting in different sensitivity to radiation. In addition, we calculated the total radiation dose (R) of tritiated water received by HaCaT cells according to this formula [38]: R = KEC_0_t. Where t is the radiation exposure time (s), which is 1.73 × 10^5^ s (24 h) in our experiment, and C_0_ is the activity of tritiated water in the medium, which is 3.7 × 10^9^ Bq/L in our experiment. E is the average energy of beta rays, 5.7 keV; K is the conversion coefficient, 1. 6 × 10^−13^ L·Gy/MeV. Thus, the total radiation dose of tritiated water received by HacaT cells in our study was 0.584 Gy.

In addition, the results of KEGG enrichment analysis and Western blot assay demonstrated that HaCaT cells treated with 3.7 × 10^9^ Bq/L tritiated water for 48 h had significant changes in up-regulating the expression of Wnt and down-regulating Jak-STAT. Some researchers have shown that the Wnt pathway is involved in embryonic development, cardiovascular, wound healing, bone regeneration and other important life activities [39,40,41,42], but its abnormal activation or mutation was usually associated with Epithelial–mesenchymal transition (EMT) processes, which is a feature of cancer development and metastasis [43]. EMT was marked by the loss of E–cadherin (CDH1) and cell–cell adhesion junctions, which Wnt could down-regulate the expression of E–cadherin [44]. Shi et al. found that ROS could induce the expression of SOX2 under hypoxia and then activated the activation of Wnt/*β*-catenin, thus promoting the EMT of HacaT cells [45]. Quan et al. proved that tritium *β*-rays could lead to increased ROS and the occurrence of an inflammatory reaction in breast epithelial cell line McF–10a cells [46]. Therefore, we hypothesized that the up-regulation of the Wnt pathway induced by tritiated water was related to ROS production. In addition, the Jak–STAT pathway is involved in the regulation of cell proliferation, differentiation, apoptosis, angiogenesis, inflammation and immune response [47]. Some drugs, such as Acitretin and Rhododendron Album Blume Extract, could inhibit the growth and invasion of HacaT cells by down-regulating the Jak/STAT pathway [48,49].

## 5. Conclusions

In this study, we found that the viability of HacaT cells decreased under tritiated water exposure, and we further analyzed the sequencing results using bioinformatics. The results demonstrated that a total of 267 DEGs, of which 191 were significantly up-regulated DEGs and 76 were down-regulated(|log2FC| ≥ 0.6, Qvalue ≤ 0.05). GO enrichment analysis showed that up-regulated and down-regulated DEGs were mainly enriched in three parts. KEGG pathway enrichment analysis showed that up-regulated DEGs were involved in Wnt and other signaling pathways, while the down-regulated DEGs were enriched in Jak-STAT and others. Finally, Wnt7b, Jak and STAT3 were selected for Western blot experiments, and the results showed that the expression of Wnt7b increased and Jak, as well as STAT3, was decreased. Thus, the sequencing results were verified by Western blot assay. The results will provide a theoretical basis for the study of tritiated water hazard mechanisms.

## Figures and Tables

**Figure 1 biology-12-00405-f001:**
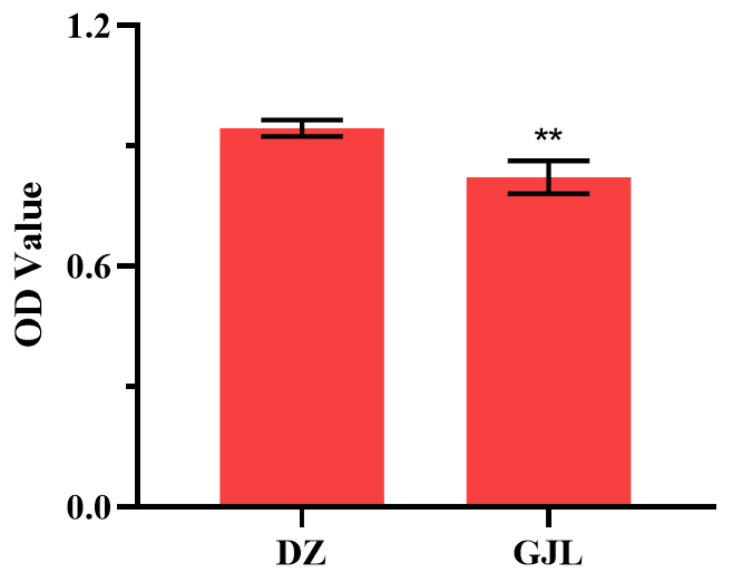
The cell viability of HacaT cells after tritium toxicity for 48 h. Data represent the mean ± SD of three independent experiments. ** *p* < 0.001; compared with the KB groups.

**Figure 2 biology-12-00405-f002:**
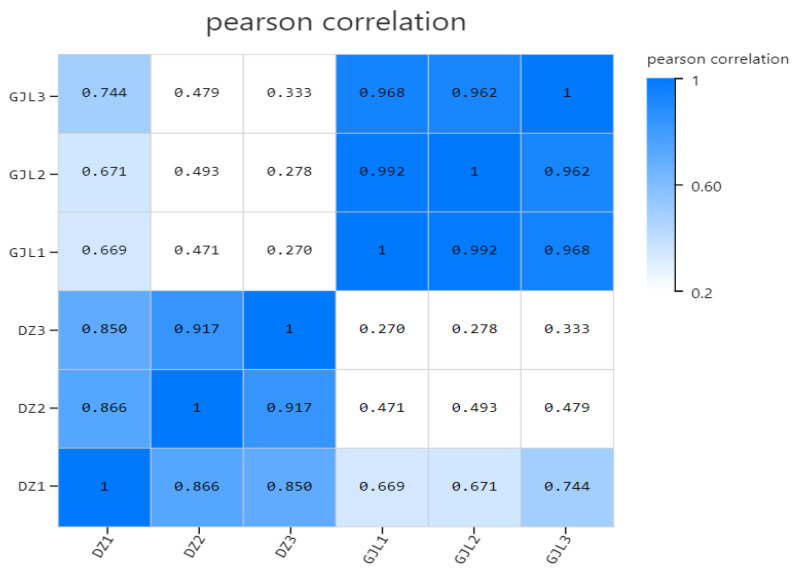
Pearson correlation heap map. The Pearson correlation coefficient is reflected in the form of a heat map. The correlation coefficient can reflect the similarity of the overall gene expression among all samples. The higher the correlation coefficient, the more similar the gene expression level is. The closer the correlation coefficient is to 1, the higher the similarity of expression patterns between DZ (DZ1, DZ2 and DZ3) and GJL (GJL1, GJL2 and GJL3) samples.

**Figure 3 biology-12-00405-f003:**
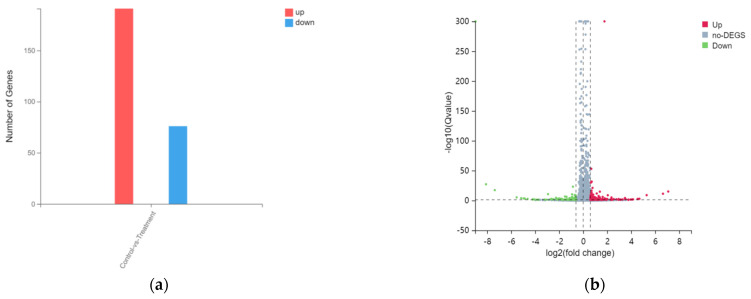
Difference analysis after tritium toxicity in HacaT cells. (**a**) Statistical map and (**b**) volcanic map of differential gene expression.

**Figure 4 biology-12-00405-f004:**
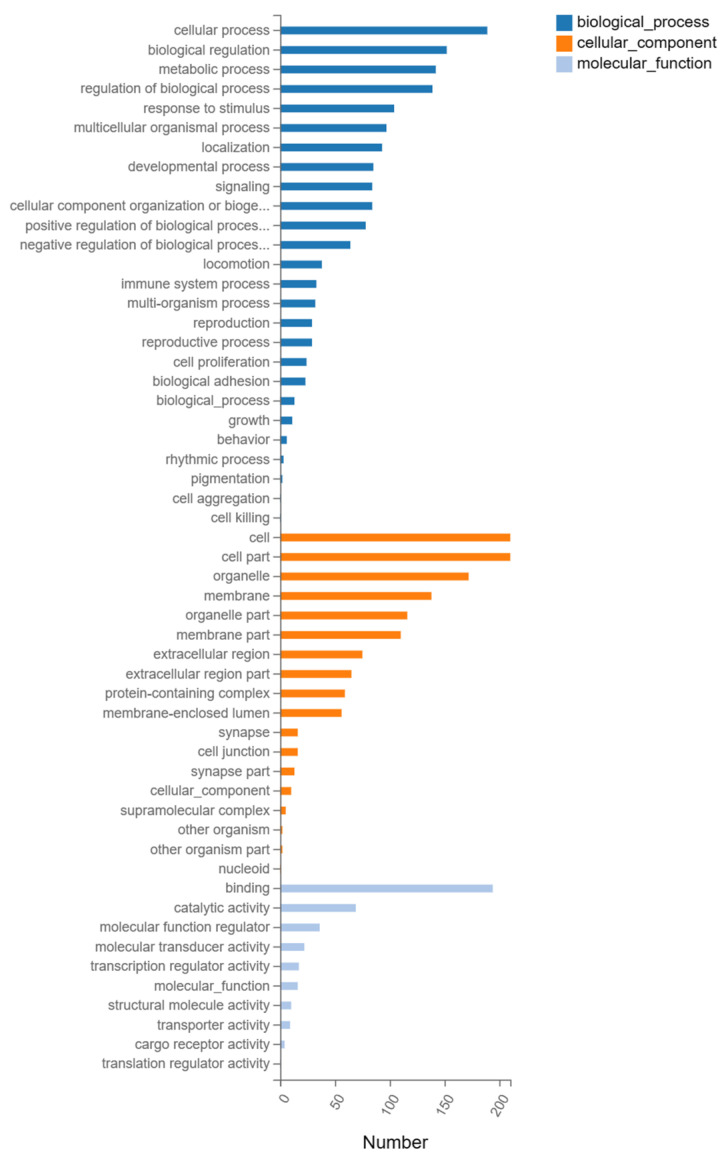
The GO classification of the total genes in HacaT cell after tritium toxicity. GJL vs. DZ of HacaT cells.

**Figure 5 biology-12-00405-f005:**
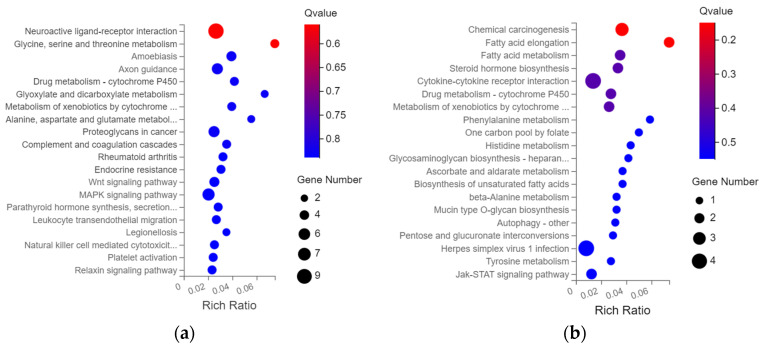
KEGG enrichment analysis. Usually, Qvalue ≤ 0.05 was considered significantly enriched. (**a**) up-regulated gene; (**b**) down-regulated gene. GJL vs. DZ of HacaT cells.

**Figure 6 biology-12-00405-f006:**
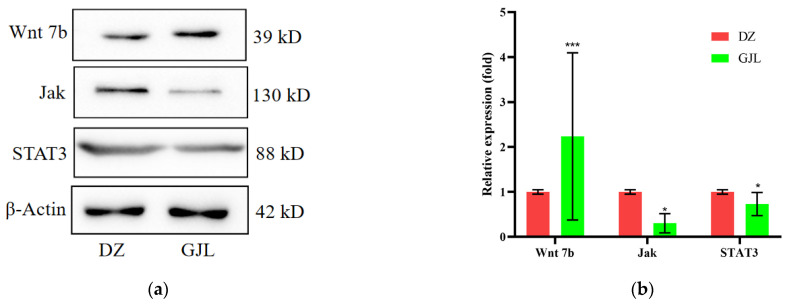
Western blot assay of differential gene expression between DZ and GJZ; (**a**) The image of protein bands in Western blot assay; (**b**) the data were quantitated and graphed. Data represent the means ± SD of three independent experiments, GJL vs. DZ of HacaT cells. * *p* < 0.05, *** *p* < 0.01; compared with the DZ groups.

**Table 1 biology-12-00405-t001:** Quality of RNA-seq data from toxicity of tritiated water samples.

Sample	Total Raw Reads (M)	Total Clean Reads (M)	Total Clean Bases (Gb)	Clean Reads Q20 (%)	Clean Reads Q30 (%)	Clean Reads Ratio (%)
DZ1	44.85	42.87	6.43	97.7	93.88	95.59
DZ2	44.65	42.99	6.45	97.54	93.48	96.28
DZ3	47.33	45.43	6.81	97.66	93.67	95.98
GJL1	47.33	44.85	6.73	97.81	94.19	94.78
GJL2	47.33	45.14	6.77	97.85	94.28	95.37
GJL3	47.33	45.17	6.78	97.58	93.55	95.45

**Table 2 biology-12-00405-t002:** Ten of the most highly enriched signaling pathways by GO enrichment analysis.

Part	ID	Description	Gene Ratio	BgRatio	*p* Value	geneID
BP	GO:0009812	flavonoid metabolic process	2/10	15/18862	2.65 × 10^−5^	445329/54576
BP	GO:0006805	xenobiotic metabolic process	2/10	120/18862	1.75 × 10^−3^	445329/54576
BP	GO:0071466	cellular response to xenobiotic stimulus	2/10	125/18862	1.89 × 10^−3^	445329/54576
CC	GO:0000782	telomere cap complex	1/11	13/19520	7.30 × 10^−3^	7015
CC	GO:0034045	phagophore assembly site membrane	1/11	15/19520	8.42 × 10^−3^	8987
CC	GO:0005697	telomerase holoenzyme complex	1/11	22/19520	1.23 × 10^−2^	7015
MF	GO:0003964	RNA−directed DNA polymerase activity	1/11	12/18337	7.18 × 10^−3^	7015
MF	GO:0001972	retinoic acid binding	1/11	20/18337	1.19 × 10^−2^	54576
MF	GO:0070034	telomerase RNA binding	1/11	22/18337	1.31 × 10^−2^	7015
MF	GO:0001223	transcription coactivator binding	1/11	26/18337	1.55 × 10^−2^	7015

## Data Availability

Not applicable.

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
