# Peer review of "Transcriptome Analysis of the Immortal Human Keratinocyte HaCaT Cell Line Damaged by Tritiated Water"

_biology, 2023, doi:10.3390/biology12030405_

Round 1
Reviewer 1 Report
Below are my critical comments:
1. “the Human immortal keratinocyte line HacaT cells” should be “the immortal human keratinocyte HaCaT cell line”.
2. "Tritium water" should be "tritiated water".
3. “tritium water may not produce external irradiation to the organism.”
Why not? What prevents it from doing that? Careful rewording is required.
4. “Radiation from tritium was harmful to living organisms.”
What doses?
Careful rewording is required. Environmental tritium such as near nuclear power plants are less than 100 Bq/L and reports about harmful effects at this level are rare to nil. Also, it has been previously shown that very-low-dose tritium levels in the environment can induce adaptive response, which is protective (not harmful), in frogs.
5. The Introduction is all about "Tritium exposure is so bad". But radiation effects are heavily dose-dependent. Plus, environmental exposure levels are very low. Nowhere in the Abstract and Introduction did the authors mention what radiation doses/radioactivity concentrations were used in the cited studies; the authors must clearly include these. Without articulation of doses/activity concentrations, biological effects of radiation are meaningless.
The Introduction fails to clearly show to readers whether what context (occupation exposure, environmental exposure, etc.) the authors were focusing on. Context dictates what dose levels are relevant to use for experimentations so that results can be meaningful. The current Introduction is not focused and lacklustre.
6. In the M&M, the authors described that they tested only one tritium activity concentration (3.7×10e8 Bq/L). In what real-life scenario do humans get exposed to this tritium level? and exposure to skin for only 2 days? What was the volume of media used in the dish? What was the volume of tritium from the stock added to media in the dish to have 3.7×10e8 Bq/L (was it 1:100 dilution?)?
Why tested only one tritium activity concentration? This is a major shortcoming of the paper.
What was the control (0 Bq/L)? Was it just cells in the growth medium? Did the authors add an equal volume of non-tritiated sterile water to this medium to count for the volume effect?
7. Why "DZ" for control and "GJL" for tritium treatment???
8. Fig 1. DZ has no SD? Please present raw OD values.
If my calculations are correct, 3.7×10e8 Bq/L should not kill HaCaT cells after 2 days of irradiation.
9. Fig 6B is questionable for stats.
Author Response
Comments from the editors and reviewers:
Reviewer: 1
- “the Human immortal keratinocyte line HacaT cells” should be “the immortal human keratinocyte HaCaT cell line”.
Reply: We greatly appreciate the reviewer’s detailed comments. According to the reviewer’s suggestion, we corrected the name of the cell in our manuscript. Please see Page 1 Line 2-3, 19-20 and Page 3 Line 104.
- "Tritium water" should be "tritiated water".
Reply: We greatly appreciate the reviewer for a meticulous review of this manuscript. According to your suggestion, we revised it in revised manuscript and correct the inaccurate words. Please see Page 1 Line 3, 17, 18, 20, 23, 26, 31, 33, 34, 44; Page 2 Line 47, 49, 50, 61, 64, 66, 68, 69, 74, 77, 80, 83, 87, 88, 91, 93; Page 3 Line 98, 99, 100, 102, 103, 109, 111, 123, 132, 133, 135, 138, 140; Page 4 Line 187, 188, 191; Page 5 Line 197; Page 6 Line 217; Page 8 Line 260, 267 and Page 9 Line 277, 278, 279, 281, 291, 296, 308, 316, 326.
- “tritium water may not produce external irradiation to the organism.” Why not? What prevents it from doing that? Careful rewording is required.
Reply: Thanks for reviewer’s kindly comment and suggestion, which is very important for us to improve our manuscript. We revised the parts in revised manuscript. Please see Page 2 Line 45-49.
Revised parts:
The average energy of tritium β—rays is 5.7 keV and maximum is 18.6 keV. It has the range of about 0.56 μm in water, which is much smaller than the average diameter of the cell (10-20 μm) [3], so tritiated water can produce biological effect on organism by irradiation.
- Kim, S. B.; Shultz, C.; Stuart, M.; Festarin, A. Tritium uptake in rainbow trout (Oncorhynchus mykiss): HTO and OBT-spiked feed exposures simultaneously. Applied radiation and isotopes. 2015, 98: 96—102. doi:10.1016/j.apradiso.2015.01.020.
- “Radiation from tritium was harmful to living organisms.”
What doses?
Careful rewording is required. Environmental tritium such as near nuclear power plants are less than 100 Bq/L and reports about harmful effects at this level are rare to nil. Also, it has been previously shown that very-low-dose tritium levels in the environment can induce adaptive response, which is protective (not harmful), in frogs.
Reply: We thank for the kindly comments of reviewer, which is very important for us to improve our manuscript. We are sorry for using such an imprecise expression. We added the part about the effects of low doses on organisms. Please see Page 2 Line 53-60. Besides, as for the damaging effect of a certain concentration of tritiated water on organisms, we give detailed answers in the reply to question 5.
Revised parts:
Low dose of tritium water may not cause damage to organisms, or even have protective effects on organisms. Stuart et al. Found that frogs exposed to tritium water radiation (5 000–35 000 Bq/L) for a long time were able to reduce their susceptibility to radiation damage [6]; Arcanjo et al. found that 0.4 and 4 mGy/h tritium water exposure enhanced the expression of involved in DNA repair, such as some genes related to DNA repair (h2afx and ddb2) increased [7]; Stuarta et al. found that B–lymphoblast cells (3B11 and FHMT–W1) were exposed to 10–100 Bq/L, and intracellular ATP content increased and cell vitality not declined [8].
- M, A, Stuart.; S, B, Kim.; D, McMullin.; A, Festarini.; T, L, Yankovich.; J, Carr.; S, Mulpuru. Adaptive response in frogs chronically exposed to low doses of ionizing radiation in the environment. J Environ Radioact. 2011, 102: 566-573.
- C, Arcanjo.; O, Armant.; M, Floriani.; I, Cavalie.; V, Camilleri.; O, Simon.; D, Orjollet.; C, A, Guillermin.; B, Gagnaire. Tritiated water exposure disrupts myofibril structure and induces mis-regulation of eye opacity and DNA repair genes in zebrafish early life stages. Aquat Toxicol. 2018, 200: 114-126.
- M, Stuarta.; A, Festarinia.; K, Schleicherb.; E, Tanb.; Sang, Bog, Kima.; Kendall, Wenb.; Jilian, Gawlikb.; Brant, Ulshc. Biological effects of tritium on fish cells in the concentration range of international drinking water standards. International Journal of Radiation Biology. 2016, 92: 563-571.
- The Introduction is all about "Tritium exposure is so bad". But radiation effects are heavily dose-dependent. Plus, environmental exposure levels are very low. Nowhere in the Abstract and Introduction did the authors mention what radiation doses/radioactivity concentrations were used in the cited studies; the authors must clearly include these. Without articulation of doses/activity concentrations, biological effects of radiation are meaningless.
The Introduction fails to clearly show to readers whether what context (occupation exposure, environmental exposure, etc.) the authors were focusing on. Context dictates what dose levels are relevant to use for experimentation so that results can be meaningful. The current Introduction is not focused and lacklustre.
Reply: The important comments are very helpful for us to improve the quality of the manuscript. After your kindly reminding, we gave a detailed description of the tritiated water concentration in our study in the introduction. In addition, we added a description of the research background of our concern in the introduction. Please see Page 2 Line 61-105.
Revised parts:
However, high doses of tritiated water radiation could damage organisms. An analysis of the chromosomes of nuclear workers who exposed to tritium (average dose 9.33 mGy) in the UK found that workers had an increased rate of chromosomal aberrations [9]. In addition, tritiated water led to liver, ovarian, leukemia, lung, breast damage, skin cancer and osteosarcoma in rats [10–12]. For example, Yin et al. proved that tritiated water above 8×1010 Bq/Mouse induced liver cancer in male rats, but only above 3.7×1012 Bq/Mouse induced ovarian cancer in female rats [10]; Seyama et al. injected 7.4×108 Bq of tritiated water into mice, which induced more than 80% of mice leukemia and lymphoma, as well as lung cancer [11]; Balonov et al. found that exposure to tritiated water in rats for 6 months at 3.7×104 Bq·g-1/Kg induced skin cancer and osteosarcoma [12]. In addtion, tritium radiation damaged the cardiovascular [13–15], immune [16,17], reproductive [18] system. Li et al. found that human B lymphocyte (AHH-1-1) was treated with 3.7×109 Bq/L for 48 h and apoptosis could be observed [17]; Lee et al. proved that micronucleus appeared in the blood samples of rats which exposed to tritiated water after 14 d in rats with a total oral dose of 3.7×104 Bq, and the appearance of micronucleus,which represents chromosome aberration. Nowosielska et al. treated mice with tritiated water of 0.888, 8.88 or 88.8×1012 Bq, and found that after 8 d, NK lymphocytes were damaged and promoted the increase of NO by macrophages [16]; Kamiguchi et al. found that after treating human sperm with 5.66–59.91×105 Bq/mL tritiated water for 80 min, the incidence of chromosome aberration in sperm structure increased linearly with the dose–dependent manner [18]. Along with some behavioral experiments showed that the learning and memory ability of mice decreased significantly when they treated with tritiated water under above 24.09×104 Bq/g after 10 d [19]. Among them, other studies showed that excessive tritium radiation also had a negative impact on the survival of algae, bacteria, shellfish and fish [20–23].
Recently, some studies have focused on the mechanism related to the decrease of cell viability caused by tritiated water. Quan et al. found that after incubation with 2×1010 Bq/L tritiated water for 3 h, the proportion of S-phase cells in human mesenchymal stem cells increased, while that of G1 and G2-phase cells decreased [24]. Vorob’eva et al. proved tritium could cause DNA damage by increasing the expression of γH2AX and dsDNA [25]. Besides, Yan et al. used 3.7×106 Bq/L tritiated water to induce the senescence of vascular endothelial cells [14], and Cui et al. demonstrated that this may be due to tritiated water down regulating the expression of c-myc via miR-34a [13]. Qiu et al. found that after treating rat nerve cells with 3.7×1010 Bq/L for 8 h reduced the expression of NCAM, which could decreased the migration ability of nerve cells [26]. Zhou et al. treated mouse brain cells with 0.19 Gy and above tritium β-rays, which resulted in increasing expression of P53 in the cells [27].
Besides, since the skin is the largest tissue in the human body, tritiated water could easily reach the skin. So, some studies have shown that tritiated water produced toxic effects on human skin cells. For example, Little et al. used a total dose of 100 Gy tritiated water to culture human skin fibroblasts for 100 h, which caused cell death [28]. However, the molecular mechanism of damage to epidermal cells caused by tritiated water has not been elucidated. Therefore, this study focused on the decrease of cell vitality caused by tritiated water to the immortal human keratinocyte HaCaT cell line (HaCaT) and related mechanisms.
- In the M&M, the authors described that they tested only one tritium activity concentration (3.7×10e8 Bq/L). In what real-life scenario do humans get exposed to this tritium level? and exposure to skin for only 2 days? What was the volume of media used in the dish? What was the volume of tritium from the stock added to media in the dish to have 3.7×10e8 Bq/L (was it 1:100 dilution?)?
Why tested only one tritium activity concentration? This is a major shortcoming of the paper.
What was the control (0 Bq/L)? Was it just cells in the growth medium? Did the authors add an equal volume of non-tritiated sterile water to this medium to count for the volume effect?
Reply: We thank for reviewer’s kindly suggestion. In practice, the human body would not be exposed to such a high concentration of tritiated water for a short time. However, the cells are different from human tissues, and the cells have been cultured in it, and the irradiation dose is a cumulative process over time. Thus, we need to take into account that if the concentration of tritiated water is reduced, the cell culture time needs to be increased. The growth of cells in the culture dish is related to the concentration of cells initially inoculated. Therefore, combined with the growth curve of HaCaT cells, our experiment controlled that HaCaT cells in the control group could grow normally within 48 h, to explore the inhibition of tritiated water on the cell viability. And the total dose of tritiated water received by cells after 48 h was reasonable compared with the actual situation.
The total volume of the medium was 5 mL, and the tritiated water was diluted at 1:10. The final concentration of tritiated water in the medium was 3.7×109 Bq/L.
Since the volume of tritiated water which we could buy for our experiment is limited, we only set one tritium treatment group (GJL group) for the experiment for considering the volume of tritiated water required by pre-experiment, MTT assay, sequencing, Western Blot assay as well as their respective repeated experiments. And we would set more experimental groups and launched further experiments.
Besides, in the DZ group, HacaT cells were cultured in 4500 μL DMEM plus 500 μL sterile water, we are very sorry for that we did not express the culture condition in the manuscript accurately. In fact, our experiment was include 3 groups: HacaT cells were cultured in 5 mL DMEM (KB group), HacaT cells were cultured in 4500 μL DMEM plus 500 μL sterile water (DZ group), HacaT cells were cultured in 4500 μL DMEM plus 500 μL tritiated water (GJL group). In the paper, we did not describe the KB group previously, and now, we correct some contents. Please see Page 3 Line 135-139.
Revised parts:
HacaT cells were treatd with 0 Bq/L and 3.7×109 Bq/L, as blank group (KB), control group (DZ) and treatment group (GJL) respectively. Specifically, the cells in KB group were cultured in 5 mL DMEM, GJL group were cultured in 4500 μL DMEM and 500 μL tritiated water which replaced with the same volume of sterile water in DZ group. A follow-up experiment was conducted after 48 h of continuous exposure to tritiated water.
- Why "DZ" for control and "GJL" for tritium treatment???
Reply: Thanks very much for reviewer’s meticulous review. In Chinese, “DZ” and “GJL” is the pinyin initials for control (dui zhao) and high dose (gao ji liang) respectively.
- Fig 1. DZ has no SD? Please present raw OD values.
If my calculations are correct, 3.7×10e8 Bq/L should not kill HaCaT cells after 2 days of irradiation.
Reply: We greatly appreciate the reviewer for a meticulous review of this manuscript. The way we processed data is to test the OD value of DZ and GJL, and then calculated the proportion of OD values decrease of GJL to DZ. The original OD values are shown in the table:
|
Groups |
OD |
||
|
KB |
0.933 0.993 0.931 |
||
|
DZ |
0.922 |
0.963 |
0.947 |
|
GJL |
0.825 |
0.861 |
0.779 |
Besides, we carefully checked the concentration of tritiated water in this manuscript. The initial concentration of tritiated water in our experiment is 3.7×1010 Bq/L, and DMEM was used to dilute tritiated water at 1:10, so the final concentration of the tritiated water should be 3.7×109 Bq/L, and we used this concentration of tritiated water to treat HaCaT cells could lead to a 13.0% decreases in the viability of HacaT cells. We are very sorry for this error for without checking carefully. We revised the concentration in revised manuscript. Please see Page 3 Line 135, Page 9 Line 278 and 296.
- Fig 6B is questionable for stats.
Reply: We greatly appreciate the reviewer for a very detailed review of this manuscript. The statistical data in Figure 6B is the ratio of gray value of each protein sample to their β-Actin average gray value, as shown in the Figure 6 (B). Please see Page 8 Line 258.
Revised parts:
Figure 6. (b) the data were quantitated
Reviewer 2 Report
Specific comments:
1. Line 10, please provide the authors’ e-mail, instead of “e-mail@e-mail.com”
2. Please be aware that Latin should be in italic style, for instance, et al.
3. Regarding M&M 2.2, could you please provide the exact concentration of tritium water?
4. Regarding M&M 2.3, I’m confused about how you quantify the dose of irradiation in tritium water and culture medium.
5. Regarding “result”, please just demonstrate the results that you have presented, do not discuss other literature. You could take apart your discussion from the results, and provide a valuable and informative discussion in the end.
6. Regarding Figures 1 and 6, Could you explain a bit about how you analyze the data? Why there is no error bar for the DZ group? And, what do DZ and GJL represent? Please provide information about biological replication and technical replication.
7. Regarding Figure 3b, there are several outstanding DEGs on both sides and top, readers might be interested in these representative DEGs and the information behind them, could you please provide the notation of these DEGs?
8. Regarding Figure 4, BothFigurese 4a and 4b are quite similar, I cannot see the differences, does it make sense? Could you please provide more explanations?
9. Regarding result 3.5, there are several signaling pathways and cellular processes that are presented in the KEGG map. Could you please explain the reason why you decide to focus on Jak-STAT signaling and Wnt signaling? And to exclude others?
10. Regarding Table 2, This table is informative, I suggest the authors visualize the information in this table or combine it with figure 4. It makes no sense to present such a large table.
Author Response
Comments from the editors and reviewers:
Reviewer: 2
- Line 10, please provide the authors’ e-mail, instead of “e-mail@e-mail.com”
Reply: Thanks for reviewer’s kindly comments. We provide the e-mail in our manuscript. Please see Page 1 Line 10.
- Please be aware that Latin should be in italic style, for instance, et al.
Reply: Thanks very much for reviewer’s meticulous review, According to your kindly suggestion, we corrected the e-mail in our manuscript. Please see Page 2 Line 45, 65, 67, 69, 72, 73, 76, 79, 87, 89, 90, 91, 92, 93, 95, 96; Page 3 Line 100; Page 4 Line 179; Page 9 Line 284, 291, 303, 305, 306; Page 10 Line 352; Page 11 Line 401, 404, 407, 447 and Page 12 Line 442, 449, 453, 456, 464.
Revised parts:
Page 2 Line 65, 67, 69, 72, 73, 76, 79, 87, 89, 91, 92, 93, 95; Page 3 Line 100; and Page 9 Line 291, 303, 306: et al.
Page 2 Line 45, 96; Page 9 Line 284, 306; Page 11 Line 401, 407 and Page 12 Line 456: tritium β–rays;
Page 2 Line 90 and Page 11 Line 404: γH2AX;
Page 4 Line 179: anti–β–Actin;
Page 9 Line 305 and Page 12 Line 442, 449, 453: β–catenin;
Page 10 Line 352: low–energy β(–);
Page 11 Line 447: TGF-β;
Page 12 Line 464: TNF–α/IFN–γ, NF–κB.
- Regarding M&M 2.2, could you please provide the exact concentration of tritium water?
Reply: Thanks very much for reviewer’s meticulous review. We used tritiated water with a specific activity of 3.7×1010 Bq/L, and added it into 4.5 mL DMEM diluted at 1:10, that is, 500 μL tritiated water, so that the final concentration of tritiated water was 3.7×109 Bq/L.
- Regarding M&M 2.3, I’m confused about how you quantify the dose of irradiation in tritium water and culture medium.
Reply: We greatly appreciate the reviewer’s detailed comments. According to formula [1] in (1), we calculated the total radiation dose Dβ of tritiated water received by HaCaT cells:
Dβ(t) =KEC0t (1)
Where, t is the exposure time (s), which is 1.73×105 s in our experiment, and C0 is the activity of tritiated water in the medium, which is 3.7×109 Bq/L in our experiment. E is the average energy of beta rays, 5.7 keV; K is the conversion coefficient, 1. 6×10-13 L Gy/MeV. Thus, the total radiation dose of tritiated water received by HacaT cells was 0.584 Gy.
- Deng, B.; Cheng, Q.; Du, Y.; Yang, Y. Cytogenetic Effects of Low Dose Tritiated Water in Human Peripheral Blood Lymphocytes. Chinese Journal of Radiological Health. 2016, 1: 6—10+14.
- Regarding “result”, please just demonstrate the results that you have presented, do not discuss other literature. You could take apart your discussion from the results, and provide a valuable and informative discussion in the end.
Reply: We thank for the kindly comments of Reviewer’s, which is essential to improving the quality of our manuscripts. Thus, we separated the discussion from the results section. Please see Page 9 Line 275-313.
Revised parts:
- Discussion
In this study, HaCaT cells were used as the experimental objects to explore the damage effects of tritiated water. We found a significant decrease in cell viability after 48 h of treatment with 3.7×109 Bq/L tritiated water. Some previously reports in the literature proved that tritiated water could also inhibit the vitality of Human Umbilical Vein Endothelial Cells and caused cell senescence after long–term exposure [13–15]. Besides, tritiated water could also affect the vitality of rat’s lymphocytes and NK cells in terms of the immune system [16,17]. As a radioactive element, the damage to cells of tritium mainly included the following two aspects: First, tritium β–decay released energy resulting in DNA single strand break (SSB) or double strand break (DSB), leading to cell apoptosis or aging [34,35]. Some characteristics related to apoptosis and senescence ensued, such as cell cycle arrest, increased γH2AX and IL-8 contents, as well as an increase in the proportion of positive cells stained with β-galactosidase (SA-β-gal) [14,24]. Second, as an ionizing radiation, Tritium caused the radiolysis of water, which promoted the formation of ROS [36] to attack intracellular biological macromolecules such as DNA, protein and lipids, causing cell damage [37]. Besides, Li et al. treated AHH-1-1 with 3.7×106 Bq/mL tritiated water, and only 72.1% of the cells survived [17]. We have similar experimental conditions, but after treatment, the HacaT cell survival rate was about 87%. This may be due to different cell types, resulting in different sensitivity to radiation.
In addition, the results of KEGG enrichment analysis and Western blot assay demonstrated that HaCaT cells treated with 3.7×109 Bq/L tritiated water for 48 h had significant changes in up–regulating the expression of Wnt and down–regulating Jak–STAT. Some researchers have showed that Wnt pathway involved in embryonic development, cardiovascular, wound healing, bone regeneration and other important life activities [38-41], but its abnormal activation or mutation was usually associated with Epithelial–mesenchymal transition (EMT) process, which was a feature of cancer development and metastasis [42]. EMT was marked by the loss of E–cadherin (CDH1) and cell–cell adhesion junctions, which Wnt could down–regulate the expression of E–cadherin [43]. Shi et al. found that ROS could induce the expression of SOX2 under hypoxia, and then activated the activation of Wnt/β–catenin, thus promoted the EMT of HacaT cells [44]. Quan et al. Proved tritium β–rays could lead to the increase of ROS and the occurrence of inflammatory reaction in breast epithelial cell line McF–10a cells [45]. Therefore, we hypothesized that the upregulation of Wnt pathway induced by tritiated water was related to ROS production. Besides, Jak–STAT pathway involved in the regulation of cell proliferation, differentiation, apoptosis, angiogenesis, inflammation and immune response [46]. Some drugs, such as Acitretin and Rhododendron Album Blume Extract, could inhibit the growth and invasion of HacaT cells by down–regulating the Jak/STAT pathway [47, 48].
- Regarding Figures 1 and 6, Could you explain a bit about how you analyze the data? Why there is no error bar for the DZ group? And, what do DZ and GJL represent? Please provide information about biological replication and technical replication.
Reply: Thanks a lot for the reviewer’s comments. For Figure 1, we obtained OD values of the cells in the DZ group and the GJL group treated with MTT through cell vitality experiment. After that, we regarded OD values in the DZ group relative to the KB group, and calculated the ratio of OD values in GJL group relative to the KB group, and then took this value as a measure of the proportion of cell vitality reduction. Now, according to your suggestion, there was error bar in Figure 1. We used the OD value to represent the difference of cell viability in Figure 1. Please see Page 3 Line 117-118 and Page 4 Line 177-181.
For Figure 6B, we obtained each protein band gray value, and calculated the ratio to that of the β–Actin of the respective group; and finally, we get the multiple of the GJL group relative to the DZ group. If we directly use the gray value to express the experimental results, the value would be too large, which affected the appearance of the picture. Therefore, we still used the the original method to describe the experimental results in Figure 6. And the gray value of the protein band was listed in the table below:
|
|
DZ |
GJL |
||||
|
Jak |
39067.894 |
31799.903 |
17575.024 |
5270.296 |
15666.439 |
5052.761 |
|
STAT 3 |
19399.317 |
20843.004 |
19485.125 |
11473.447 |
13923.983 |
20670.296 |
|
Wnt |
4491.418 |
11694.903 |
18848.711 |
24200.388 |
11430.146 |
25661.61 |
|
β–Actin |
43154.563 |
49460.3 |
|
53087.806 |
45886.2 |
|
Revised parts:
2.2 Cell Viability Assay
HacaT cells were grown to 70% of the culture surface area after cells were attached to the wall and fully stretched, and exposed by different concentrations of tritiated water, and then cells were cultured for 48 h. The cell viability was launched with MTT (Sigma–Aldrich, MO, USA), and the OD value was measured at 492 nm. Then, the value of cell viability was measure by the change of OD value.
The results of MTT experiment found that tritiated water could lead to the viability of HacaT cells significantly decreases as shown in Figure 1. The OD value of the DZ group in MTT experiment was about 0.94, while that of the treatment group was about 0.82, which was 13.0% lower than that of the DZ group. Thus, tritiated water could impair the viability of HacaT cells significantly.
Figure 1. The Cell viability of in HacaT cells after tritium toxicity for 48 h. Data represent the mean ± SD of three independent experiments. *** p < 0.001; compared with the KB groups.
- Regarding Figure 3b, there are several outstanding DEGs on both sides and top, readers might be interested in these representative DEGs and the information behind them, could you please provide the notation of these DEGs?
Reply: Thanks for reviewer’s kindly and thorough suggestion. We added the details of several prominent DEGs on the sides and top, and we put the detailed information about these genes in the SI Table 1.
Revised parts:
SI Table 1. DEGs on the sides and top in volcanic map
|
Gene ID |
Gene Symbol |
log2 FC |
-log10 (Q value) |
category |
|
107987457 |
LOC107987457 |
-8.09 |
27.20 |
Down |
|
100526740 |
ATP 5MF-PT CD1 |
-7.36 |
17.41 |
Down |
|
100526832 |
PHOSPHO2-KLHL23 |
7.09 |
15.09 |
Up |
|
54921 |
CHTF8 |
6.65 |
11.37 |
Up |
|
56100 |
PCDHGB6 |
5.29 |
8.98 |
Up |
|
6818 |
SULT1A3 |
1.77 |
300.00 |
Up |
- Regarding Figure 4, Both Figures 4a and 4b are quite similar, I cannot see the differences, does it make sense? Could you please provide more explanations?
Reply: We greatly appreciate the reviewer’s kindly comment. There is little difference between up and down-regulated genes in the GO classification. Thus, we have reanalyzed the GO classification of the total (up+down) genes. Please see Page 6 Line 237-247 and Page 7 Line 248-252 and SI Table 2.
Revised parts:
3.3. GO enrichment analysis of DEGs
GO enrichment analysis wass divided into 3 parts: molecular function, biological process and cellular component. Figure 4 showed the GO enrichment analysis in the total DEGs between GJL and DZ group in the terms of biological process module, DEGs enriched in cellular process, biological regulation, metabolic process, regulation of biological process and so on. In the terms Cellular Component, DEGs enriched in cell, cell part, organelle, membrane and so on. While in the terms of binding, catalytic activity, molecular function regulator, molecular transducer activity and so on. Specifically, we list 10 of the most highly enriched signaling pathways according to P value in Table 2. Besides, all the specific genetic information of DEGS analyzed by GO was listed in detail (SI Table 2).
Figure 4. The GO classification of the total genes in HacaT cell after tritium toxicity. GJL vs DZ of HacaT cells.
Table 2. Ten of the most highly enriched signaling pathways by GO enrichment analysis
|
Part |
ID |
Description |
Gene Ratio |
BgRatio |
pvalue |
geneID |
|
BP |
GO:0009812 |
flavonoid metabolic process |
2/10 |
15/18862 |
2.65E-05 |
445329/54576 |
|
BP |
GO:0006805 |
xenobiotic metabolic process |
2/10 |
120/18862 |
1.75E-03 |
445329/54576 |
|
BP |
GO:0071466 |
cellular response to xenobiotic stimulus |
2/10 |
125/18862 |
1.89E-03 |
445329/54576 |
|
CC |
GO:0000782 |
telomere cap complex |
1/11 |
13/19520 |
7.30E-03 |
7015 |
|
CC |
GO:0034045 |
phagophore assembly site membrane |
1/11 |
15/19520 |
8.42E-03 |
8987 |
|
CC |
GO:0005697 |
telomerase holoenzyme complex |
1/11 |
22/19520 |
1.23E-02 |
7015 |
|
MF |
GO:0003964 |
RNA-directed DNA polymerase activity |
1/11 |
12/18337 |
7.18E-03 |
7015 |
|
MF |
GO:0001972 |
retinoic acid binding |
1/11 |
20/18337 |
1.19E-02 |
54576 |
|
MF |
GO:0070034 |
telomerase RNA binding |
1/11 |
22/18337 |
1.31E-02 |
7015 |
|
MF |
GO:0001223 |
transcription coactivator binding |
1/11 |
26/18337 |
1.55E-02 |
7015 |
- Regarding result 3.5, there are several signaling pathways and cellular processes that are presented in the KEGG map. Could you please explain the reason why you decide to focus on Jak-STAT signaling and Wnt signaling? And to exclude others?
Reply: Thanks very much for reviewer’s meticulous review. Firstly, we found 267 DEGs by sequencing, so we could not verify all of them but we picked Jak/STAT and Wnt. Both of them were classical cell signaling pathways, which involved in many important life activities, such as cell growth, proliferation, differentiation, migration, apoptosis and so on. Secondly, we also need to combine the existing antibodies in the laboratory, or whether we can buy them, as well as we should consider the price, and whether the volume of tritiated water used in the experiment is enough for Western Blot experiment. For the above reasons, we selected 3 proteins on these two pathways for experimental verification.
- Regarding Table 2, This table is informative, I suggest the authors visualize the information in this table or combine it with figure 4. It makes no sense to present such a large table.
Reply: Thanks for reviewer’s kindly detailed comment, which was very helpful for us to significantly improve the manuscript quality. We simplified Table 2, and put those contents into SI Table 2. Please see Page 7 Line 251-252.
Revised parts:
Table 2. Ten of the most highly enriched signaling pathways by GO enrichment analysis
|
Part |
ID |
Description |
Gene Ratio |
BgRatio |
pvalue |
geneID |
|
BP |
GO:0009812 |
flavonoid metabolic process |
2/10 |
15/18862 |
2.65E-05 |
445329/54576 |
|
BP |
GO:0006805 |
xenobiotic metabolic process |
2/10 |
120/18862 |
1.75E-03 |
445329/54576 |
|
BP |
GO:0071466 |
cellular response to xenobiotic stimulus |
2/10 |
125/18862 |
1.89E-03 |
445329/54576 |
|
CC |
GO:0000782 |
telomere cap complex |
1/11 |
13/19520 |
7.30E-03 |
7015 |
|
CC |
GO:0034045 |
phagophore assembly site membrane |
1/11 |
15/19520 |
8.42E-03 |
8987 |
|
CC |
GO:0005697 |
telomerase holoenzyme complex |
1/11 |
22/19520 |
1.23E-02 |
7015 |
|
MF |
GO:0003964 |
RNA-directed DNA polymerase activity |
1/11 |
12/18337 |
7.18E-03 |
7015 |
|
MF |
GO:0001972 |
retinoic acid binding |
1/11 |
20/18337 |
1.19E-02 |
54576 |
|
MF |
GO:0070034 |
telomerase RNA binding |
1/11 |
22/18337 |
1.31E-02 |
7015 |
|
MF |
GO:0001223 |
transcription coactivator binding |
1/11 |
26/18337 |
1.55E-02 |
7015 |
Round 2
Reviewer 2 Report
Thanks for considering my suggestions. This paper is generally better than the last version. I am personally satisfied with it.